# The Path towards Predicting Evolution as Illustrated in Yeast Cell Polarity

**DOI:** 10.3390/cells9122534

**Published:** 2020-11-24

**Authors:** Werner Karl-Gustav Daalman, Els Sweep, Liedewij Laan

**Affiliations:** Department of Bionanoscience, TU Delft, 2629 HZ Delft, The Netherlands; W.K.Daalman@tudelft.nl (W.K.-G.D.); E.Sweep@tudelft.nl (E.S.)

**Keywords:** network evolution, *Saccharomyces cerevisiae*, polarity, modularity, neutrality, symmetry breaking

## Abstract

A bottom-up route towards predicting evolution relies on a deep understanding of the complex network that proteins form inside cells. In a rapidly expanding panorama of experimental possibilities, the most difficult question is how to conceptually approach the disentangling of such complex networks. These can exhibit varying degrees of hierarchy and modularity, which obfuscate certain protein functions that may prove pivotal for adaptation. Using the well-established polarity network in budding yeast as a case study, we first organize current literature to highlight protein entrenchments inside polarity. Following three examples, we see how alternating between experimental novelties and subsequent emerging design strategies can construct a layered understanding, potent enough to reveal evolutionary targets. We show that if you want to understand a cell’s evolutionary capacity, such as possible future evolutionary paths, seemingly unimportant proteins need to be mapped and studied. Finally, we generalize this research structure to be applicable to other systems of interest.

## 1. Introduction

How cells work and how they evolve is at the heart of cell biology. In this work we will review how cellular architecture (“how cells work”) and its evolutionary properties (“how they evolve”) are related to each other. Understanding evolution and possible mutational paths of protein networks, and especially the cell polarity network, is not only satisfying our curiosity but may also help us understand and possibly predict cancer progression [1].

Every cell consists of many different interconnected functional protein networks (for definitions, see Table 1), such as transcription, translation, or polarity establishment [2]. The network’s architecture, (for example: which protein binds to/reacts with which other protein), impacts the evolutionary possibilities of a network in multiple ways. For example, hubs, proteins with many binding partners, tend to evolve slower [3]. Less connected proteins, that may be deleted in a cell without a detectable change in cell physiology, can permit duplication of other genes and thus promote evolution [4]: duplicates of a gene enable new options for diversification, which facilitate further evolution of a gene/protein and the surrounding network [5,6,7]. Interestingly, many mutations (from 3% of nonsilent mutations in bacteria to 30% in hominids [8]) in a cell show very weak, or no effect on the cell’s function, a phenomenon called neutrality [9]. Thus, proteins that may seem unimportant for how the cell works now, in this environment, may become important when changes occur in the network architecture due to a mutation or a switch in environment [10,11].

A well-studied model organism to concretize how these proteins, without a detectable phenotype, shape a network is *Saccharomyces cerevisiae*, or budding yeast. The organism generally exhibits many of the network properties defined in Table 1, such as hierarchy and the presence of hubs [2]. Here, neutrality is also pervasive, as only 40% of homozygous gene deletions for the entire organism initially had obvious phenotypes [12]. Moreover, the environment has been shown to have a notable influence on neutrality, as lethal heterozygous deletions can be compensated by poor medium [13]. As a general rule, both the network architecture and the environment can mask the function of many proteins.

Within this organism, a, to some extent, representative example of a protein network is the polarity network, which governs how the yeast chooses a direction in which to divide and involves directing dozens of proteins in a process of breaking its internal spherical symmetry (see, e.g., [14]). As required, we observe the presence of the common network properties demonstrated in Section 2, such as hierarchy and redundancy. Polarity is also one of many biological functions in yeast for which a subdivision of many proteins into a quasi-modular network proved possible [15]. Within polarity, even more detailed submodules can be distinguished [16]. Neutrality is also exhibited by several polarity proteins discussed in Section 3, in part responsible for the difficulty in determining the role of each protein. Lastly, polarity is a pattern formation process where, by definition, spatiotemporal dependencies are important, and understanding evolution generally relies on understanding this type of dependencies [17].

However, the polarity network is not a prime example of the sort of networks with abundant transcriptional regulation. Other templates are better suited for learning about the evolution of gene regulation, such as interaction networks centered around transcriptional regulators (e.g., Mcm1 [18]), ribosomal regulation (e.g., [19]) the stress response (e.g., [20]) or metabolic response (e.g., *GAL* pathway [21]). The existence of established regulatory templates thus conveniently complements our focus on symmetry breaking during polarity as a model for the protein interaction network, which is also a topic for evolutionary studies.

Concretely, in [22] a mutant strongly defective in polarity establishment was experimentally evolved and found to recover remarkably reproducibly, e.g., the first rescuing mutation to sweep the population was always the same. Because of this exhibited tractability of the adaptations, network structures within the polarity network that facilitated evolution could be concretely interpreted in terms of redundancies [23]. In another approach to determine the flexibility of the polarity network, historical evolution was studied for 40+ proteins in almost 300 fungal species in [11]. Again, the polarity network exhibited sufficient modularity so that studying its evolution separately from other functions still yielded interpretable results. For example, authors showed that polarity network size was shaped in part by fungal lifestyle (e.g., uni- or multicellular).

Nevertheless, clear justification for observed evolutionary trajectories in this network remain difficult to make. It is sometimes possible to generate more abstract predictions by linking network architecture to the evolvability of the polarity network using classical regulatory motifs. For example, the presence of positive and negative feedback in polarity establishment confers robustness to the network [24,25], which in turn may facilitate evolution. But in order to make more concrete and detailed predictions about evolutionary trajectories, an important insight is that we need mechanistic information of (parts of) the polarity network.

To illustrate this point, a bottom-up model was constructed in [26] where molecular details were coarse-grained following analysis on numerical simulations of multiple polarity mechanisms [23]. This approach was successful in quantitatively describing the fitness along the evolutionary trajectory in [22]. Furthermore, the predictions on epistasis, an important bottleneck for predicting evolution [27], can be extended to other modules, and although use of full mechanistic understanding is superior in quantitative assessments of epistasis, biofunctional information (viz. from GO-terms, in agreement with [28]) as input to the model suffices for epistasis sign predictions.

Instructive is the Nrp1 case where full information is absent, but some phenomenological information is available. Based on the latter, inclusion of Nrp1 into the bottom-up coarse-grained model of [26] is still worthwhile, but leaves room for improvement. This marks the importance of continuing to investigate protein networks until molecular mechanisms have been elucidated. In this review we advocate that obtaining this knowledge is (soon) feasible, motivating the use of the yeast polarity system for studies in network organization (with properties such as hierarchy, modularity/connectivity and redundancy that can “hide” proteins) as well as evolution.

We present three case studies in our review to reflect different stages in this knowledge quest; while literature on the first case, bud scar proteins, has been quite advanced for several years, only recently the GAP mechanism [23] (second case) has been revealed and for Nrp1 (third case) more work remains. These cases illustrate how to move from detecting the strong and more obvious phenotypes to unveiling evolutionary important hidden features (such as for Nrp1). The cases build on literature of the polarity network summarized in the form of a Venn diagram, ideal for depicting a hierarchical and semimodular protein grouping. The feasibility of deciphering a protein’s role in this Venn diagram turns out to depend on how deep the protein is embedded inside the network, which can impose neutrality on mutations to hinder unique attribution of genotypes to phenotypes. In the sections thereafter, our three examples with varying depths of embedding serve to show how improved understanding of the nontrivial parts of the networks can elucidate evolutionary trajectories. Ultimately, we believe that in all the work done in yeast for many decades by many researchers there is a common recipe applicable and useful for many protein networks, as expanded upon in the outlook.

## 2. Polarity Overview

Within the yeast polarity network, four pathways to polarization exist which cannot easily be considered modular. Their interconnectivity can be conveniently visualized in the form of a Venn diagram (Figure 1, with references found in Appendix A
Table A1). These pathways are hierarchically set up, in the following order: the mating pathway at the top, then the bud scar pathway, followed by the reaction−diffusion pathway and finally the actin pathway. In short, their function boils down to the act of condensing the GTPase Cdc42 bound to GTP molecules (i.e., active Cdc42) to one point on the plasma membrane, which can signal downstream effectors to proceed the cell cycle [29]. To prevent premature or overdue localization of active Cdc42 and allow some influence on the hierarchy of pathways, a fifth pathway exists to control the previous four, namely the timing pathway. The next section summarizes the most important interactions in and across all pathways, starting with the timing cue, before expanding upon the three examples. As a site note, we have done our best to include all relevant papers, but apologize for important papers we have missed.

### 2.1. Timing: The Control Knob

During isotropic growth in G1, active Cdc42 localization is suppressed by overactivity of its associated GTPase activating proteins (GAPs) and sequestration of its guanine nucleotide exchange factor (GEF). The consequence of both circumstances is the vast abundance of inactive Cdc42, which is bound to GDP instead of GTP [30], rendering it impossible to signal the polarity cue. The purpose of this pathway (see top dark purple region in Figure 1) is hence to timely reduce GAP activity and release the GEF, which must be in response to important physiological parameters that indicate the readiness of the cell: sufficient protein production, a sufficient size, and sufficient nutrition.

The physiological state of the cell enters the equation through nuclear levels of cyclin Cln3. Upon sufficient nutrition and size, Cln3 levels rise either more directly through higher Cln3 mRNA abundance [31], or more indirectly through Ydj1 disturbing Cln3 localization by Whi3 [32], the latter also being an inhibitor of Cln3 mRNA translation [33]. The arrival of nuclear Cln3 allows binding partner and cyclin-dependent kinase Cdc28 [34] to phosphorylate Whi5, which had inhibited expression of Cln2, another cyclin [35,36]. Cln2 can then reinforce its own expression, consolidating the original Cln3 signal [37].

Now, the Cdc28-Cln2 complex can distribute the physiological signal to the aforementioned targets, the GAPs and the GEF. The kinase Cdc28 phosphorylates all four GAPs Bem2, Bem3, Rga1 and Rga2 [38,39,40,41] and Far1, which was keeping the GEF Cdc24 in the nucleus [42]. Now cytoplasmic levels of active Cdc42 can rise, leading to polarity establishment through subsequent pathways.

Importantly, the completion of the timing pathway causes the hierarchy of the subsequent pathways to change. While the mating pathway is otherwise dominant, the kinase Cdc28 phosphorylates Ste5, a crucial hub in the mating pathway, to stop the mating in its tracks [43,44,45]. In the following discussion of the mating pathway, the situation is considered where the timing pathway did not overwrite its behavior.

### 2.2. Mating: Heavily Cross-Linked

The mating pathway is the dominant force across the four symmetry-breaking pathways. While polarization in a random orientation is possible after the timing cue (see the section on reaction−diffusion further on), the presence of pheromones of the opposite mating type (a or α) should redirect the Cdc42 localization to the side of the pheromone signal. This process revolves around Ste5, as also depicted in the left, blue-grey circle of Figure 1.

Briefly put, once pheromones bind membrane proteins Ste2 and Ste3 [46,47], Ste4 is released from the membrane [48,49] and binds Ste20 and scaffold Ste5. This scaffold binds Ste7, Ste11 and Fus3, which are activated by sequential phosphorylation [50,51,52]. Fus3 may inhibit the GAPs Bem2 and Bem3 [39], while Ste5 binds the GEF Cdc24 [53], replacing the absence of the timing pathway result to stimulate activity of Cdc42.

While this simplified view would suffice to redirect the Cdc42 localization, the mating pathway is much more intertwined with the other pathways than seemingly necessary, particularly with the actin pathway. The abundant mechanistic redundancies result in a more complex picture, obfuscating the role of the proteins involved. For example, active Cdc42 stimulates Ste11 phosphorylation/activation [54]. Another form of positive feedback, as well as a bridge to the actin pathway, is the Cdc42 recruitment of formin Bni1 [55]. The resulting nucleation of actin cables may transport Ste5-GEF Cdc24 complexes [56], possibly also through Bem1. This scaffold co-immunoprecipitates with Act1 [57] and Far1 [58], which is bound to the GEF Cdc24 [59], but is itself in turn also bound to Bem1 [60]. Another actin cross-link is the phosphorylation and localization of Bni1 through Fus3 [61]. Clearly, care must be taken in assigning roles to different proteins, as many are overloaded.

### 2.3. Bud Scar: Mostly Modular and Ordered

In the absence of a mating cue, the timing pathway reduces GAP activity and releases the GEF, while the mating pathway is repressed. Under the new hierarchy, the bud scar pathway is normally dominant. The scar refers to leftover proteins from the previous division, named septins [62,63]. This spatial cue can be exploited for polarity establishment; a new bud forms adjacent to the scar (axial budding, haploids) or also at the opposing side (bipolar budding, diploids) [64,65]. The bottom, dark blue circle of Figure 1 represents this path from septins to Cdc42 recruitment graphically. More background information about the core bud scar protein group Bud1 to Bud5 is discussed separately in one of the three case studies, and only a brief overview of the pathway as a whole is discussed in this section.

An important bud scar localization target is Bud5, which activates and recruits Bud1 [66,67], a protein also known as Rsr1 [65,68]. In haploids, where Axl1 is specifically expressed [69], the Bud5 localization occurs by relay of the septin signal to a protein complex of Bud3, Bud4, Axl1, Axl2 and Bud5 [70,71,72,73]. In diploids, functionality of Rax1 and Rax2 is not impaired, presumably by blocked expression of Axl1 [74], so these can localize Bud9 and Bud8 to the bud scar or the opposite end of the scar respectively [75], which in turn recruit Bud5 [76].

After Bud5, localization follows of Bud2 [77], the GAP for Bud1 [78], to complete the control of the GTPase cycle of Bud1. Finally, Bud1 binds GEF Cdc24 and Bem1 [79] (although Cdc24 has the strongest affinity with Bem1 [80]), to redirect the pattern formation made possible after the timing cue. As linkage of Cdc42 GAP Rga1 to septins prevents reuse of the previous location [81], the new bud forms adjacent to the bud scar.

As a whole, the bud scar pathway is not completely modular either. Aside from nudging the reaction−diffusion pathway (see next section), an example of a cross-link is that Bud8 and Bud9 are delivered by actin transport [82]. The highest position in the hierarchy in the absence of the mating cue is also not absolute; multiple ways to promote the subsequent reaction−diffusion pathway exist, such as deletion of Bud1 and Bud8 [83], Axl2 and Rax1 [84], or Bem1 [22]. Therefore, it has been possible to retrieve the information discussed in the following section.

### 2.4. Reaction−Diffusion: Ample Redundancy

Even in the absence of chemical or spatial cues, the shift in balance towards activation of Cdc42 induced by the timing pathway still provides the conditions for swift symmetry breaking. Theoretical models concerning this pathway have been subject to updates as more molecular details have been revealed. Central is the strong positive feedback generated by the Bem1-GEF Cdc24 complex, as modelled in, e.g., [85,86] with further refinement in [87]. This feedback is sufficient for polarity success, which becomes rather insensitive to GAP abundance. More details on the GAPs were uncovered in [23] and are placed in a broader context in the case study further on.

What makes this pathway special is the limited number of proteins that are unique to this pathway, as seen from the central, emerald circle in Figure 1, namely only Cla4 and Rdi1. The latter is the least cross-linked of the two, providing a possible justification for referring to the WT mechanism as the Rdi1 polarity mechanism, as in [24]. Cla4 is more context-dependent, possibly having two opposing roles, promoting and inhibiting polarity [23,88,89]. Yet both Rdi1 and Cla4 are dispensable for polarity [88,90].

An even stronger addition to the redundancy within this pathway is on the positive feedback side. Without Bem1, generic rescuing feedbacks suffice [23], among which Cla4 could account for 20% of their function [91]. More feedbacks may be found in the GAPs (see GAP case study) through actin transport as described in, e.g., [24]. This brings us to the actin pathway as the final layer to discuss.

### 2.5. Actin: The Mysterious Auxiliary Layer

The actin pathway (rightmost green circle in Figure 1) has featured several times already in the previously discussed pathways, but its individual role is still quite uncertain. Yeast formin Bni1 which nucleates actin cables, binds active Cdc42 [55], and is known to be involved in exo- and endocytosis [92,93]. This suggests transport of polarity proteins from and to the presumptive bud site. The resulting actin pathway has been confusingly implicated in two opposing roles; promoting Cdc42 polarization, see, e.g., [24,94,95], as well as negatively impacting Cdc42 polarization [96,97].

A way to reconcile these findings is that actin transport contributes to a process promoting Cdc42 polarization but without relying on significant transport of Cdc42 itself. As mentioned in the mating pathway, Bem1 and Act1 co-immunoprecipitate [57], suggesting that Bem1, and concordantly its multiple binding partners, might get transported through the actin pathway. However, in the absence of Bem1, 80% of the positive feedback is still unidentified [91], which may very well be actin-related. Instead, a prime candidate is the GAP group, which is known to bind the epsin-coating of actin cables involved in endocytosis [98]. This is further discussed in the case study on the GAPs. In any case, it is quite difficult to decipher the actin pathway, in large part due to its low positioning in the yeast polarity hierarchy.

## 3. Case Studies

In the introduction the need for determining the (potential) functions of a protein inside a complex network even when these are normally hidden, was explained as, evolution may exploit these later. In the following sections, the reconstitution of the mechanistic details involving three protein(s) (classes), that are currently at different stages in their discovery (see graphical mechanistic summary in Figure 2), is illustrated; the Bud1-5 bud scar protein subset, the GAP proteins for Cdc42, and finally Nrp1.

In the first case, detailed knowledge has been established the longest, and has also been put in an evolutionary perspective as forming a “weak regulatory link” that facilitates phenotypic variation [99,100]. Therefore, relatively small gains are expected from the remaining open questions. For the GAP proteins, detailed knowledge is only very recently obtained. Together, the purpose of putting the literature of these cases into historical context is that these two examples form convenient templates to delineate the route towards complete understanding for proteins as in the final case, Nrp1, where knowledge is scarce. On the one hand, technical advances that improve detection of phenotypes were the most prominent method of progress for the bud scar proteins. On the other hand, the GAP protein and the Nrp1 case show that progress can rely more on very specific designs of experiments than on new technology. All three cases combined in turn illustrate the foundation under the general strategy to approach networks put forward in the outlook.

### 3.1. Bud Scar Proteins

One of the strongest phenotypes to observe for budding yeast is the location of the next bud. Already more than half a century ago, it was documented that *S. cerevisiae* exhibited two possible budding patterns [64]. Normally for haploids, the next bud grows next to the previous division site (axial budding), whereas diploids can also pick the location opposite the previous site (bipolar budding). Changes in this pattern are clear phenotypes, and are therefore a useful detection tool in bulk mutagenesis screens. In 1991, five genes were identified that affected (seemingly as their sole phenotype) the bud site selection, and were therefore named *BUD1* (formerly known as *RSR1* [68]), *BUD2*, *BUD3*, *BUD4* and *BUD5* [65,66]. These genes and their associated proteins are therefore straightforwardly localized in the bud scar circle in the Venn diagram (Figure 1).

The next step, retrieving information on physical interactions, is more elaborate. For example, authors in [66] reasoned the physical mechanism of Bud5 (amongst others a GEF for Bud1) based on sequence information, which was further substantiated in [78], together with the GAP action of Bud2. Bud2 and Bud5 hence underlie the relevant GTPase cycling of Bud1, as in vitro studies established that Bud1 when GTP-bound, mostly binds Cdc24, while binding and recruiting Bem1 when GDP-bound [79,101]. This provides the bridge to the previously described reaction−diffusion pathway. With the advent of GFP [102], it also became possible to get in vivo spatiotemporal information from yeast proteins, later even in bulk [103]. Now armed with the tool of fluorescence microscopy, substantial progress was made for validating the roles for Bud2 and Bud5 [77].

The final confirmation on the mechanisms comes when zooming into the domains of a protein, by expressing truncated versions or intelligently chosen point mutations. For example, the domains responsible for how Bud5 affects whether budding is axial or bipolar are described in [104]. For other proteins, however, it was necessary to consider in greater detail the protein network that the bud genes compose. Therefore, it has taken more time to reverse-engineer the details for Bud3 and Bud4. Authors in [72] combine their findings with earlier literature to demonstrate that septins from the previous division localize Bud4, which can then bind Bud3. Subsequently, Bud4 binds Axl2, and after this Axl1 as well. Domain analysis of Bud3 further uncovered more details [105] to show Bud3 can serve as a GEF for Cdc42 (but early in G1, before Start), which is important for the Bud4−Axl1 link.

This leaves only a few mechanistic unknowns. For example, the exact recruitment of Bud2 as depicted in Figure 2 is putative, as it is unclear how Bud2 is recruited even in the absence of Bud1 or Bud5 [74]. Bud1 can recruit itself by dimerization and Cdc42 at the bud site of the membrane [67]. In the same paper, it is shown that while Bud5 also recruits Bud1, the former may only indirectly be involved in the Bud1 dimerization, considering the full GTP-GDP cycle is critical for appropriate dimerization. This suggests guidance of Bud2 for the self-recruitment, but the exact order for Bud1-binding and hydrolysis of its GTP is open for interpretation, given the alternative model in [74].

As an alternative representation of the current state of research of the bud scar proteins, Figure 3 depicts an ordering in the form of a mind map. For clarity, only Bud1 is considered. As can be seen, the bud scar protein case is relatively well advanced, with ample coverage of all categories.

### 3.2. GAPs

Central to the function of GTPase Cdc42 are proteins that promote GTP hydrolysis, i.e., its GTPase activating proteins. Bem3, was the first to be identified [106], followed by Rga1 and Rga2 [107,108,109]. A summary of Bem3 studies/information is given in Figure 4. The specific molecular function of GAPs allowed validation in vitro, where Cdc42-GTP was incubated with a GAP to determine whether the amount of GTP indeed decayed faster than without a GAP. In [106] Bem2 was not found to exhibit detectable GAP activity, and not until [110] could Bem2 be convincingly considered a GAP as well.

After their establishment as a GAP, the localization of Bem3, Rga1 and Rga2 was determined using either antibody staining or GFP-tagging [111]. These GAPs colocalized with Cdc42 at the bud site (although Rga1 is slightly more dispersed). Bem3 was also found in Spitzenkörper-like structures, but seemingly after polarity establishment during polarized growth [112]. The GAP localizations affirmed their role in polarity, although they also seemed somewhat redundant, as a triple mutant was still viable [109].

More information on GAPs was gathered through their interactions. Bem2 and Bem3 were found to be associated with the mating pathway [39], Bem3, Rga1 and Rga2 related to the actin pathway [98,113], while [81] suggested a link between Rga1 and the bud scar pathway. However, several open questions remain. For example, why is there more than one GAP? Why and how are these distributed across multiple pathways?

More clues were to follow from the tedious deciphering of the mechanistic action of the GAPs. The difficulty behind this problem was clearly elucidated by the model of [87]. There, authors showed the dominance of the Bem1-mediated positive feedback minimized the phenotypical influence of varying GAP concentration. Consequently, this led to the realization that GAP details only emerged in a Δ*bem1* background.

In [23] a proposed mechanistic model for the GAPs was validated against this background. The idea was that GAPs were temporarily retained on the membrane by Cdc42 during the GTP hydrolysis process. Only in the location in the cell with high membrane concentrations of Cdc42-GTP would this lead to a local depletion of available GAPs, which cannot be compensated by the cytosolic diffusive flux. Elsewhere on the membrane, this was not a problem and Cdc42 was promptly inactivated and recycled, leading to only one spot on the plasma membrane where active Cdc42 accumulated. A generic positive feedback mechanism for Cdc42 (due to, e.g., Cla4) completed the symmetry breaking. This provides a good example of the added value of revealing nonobvious mechanistic details. This GAP model, which is obscured in the presence of Bem1, allowed the authors in [23] to provide a detailed explanation of the evolutionary trajectory observed in [22].

Yet, this may not be the complete role of the GAPs. Their mechanism partly supplements the need for the Bem1-mediated positive feedback, but a further deletion of *CLA4* and *BEM3* reveals that 80% of the positive feedback originates elsewhere [91] (p. 101). Here the interconnectivity with another pathway can surface, which initially seemed elusive and redundant.

As discussed in the actin pathway, there could be a critical link with actin and the GAPs. The aforementioned local depletion of unbound GAPs may be reinforced by actin transport [91] (p. 34). Epsin coatings of endocytic vesicles colocalize with polarized growth and bind GAPs [98,113]. Moreover, active Cdc42 releases the auto-inhibition of kinases Ste20 [54] and Cla4 [114], both of which phosphorylate myosins 3 and 5 to ultimately lead to activation of the Arp2/3 complex [115], critical for endocytosis. In this way, sites of active Cdc42 can promote the endocytosis which may reinforce the stability of the site, providing the feedback needed to establish polarity (model F in [116]). The combination of interactions of recyclable GAPs and active Cdc42 would also fulfil the requirement of actin-mediated recruitment formulated in [117], provided the GAPs diffuse slowly on the membrane [96].

Figure 4 graphically summarizes the available information for one of the GAPs, Bem3. However, there is still room for improvement with experiments whose design can just now be established. As with the bud scar pathway, subtle information may be retrieved through experiments at the domain level, to test, e.g., the GAP trafficking hypothesis. One could remove the link between GAPs and actin through deletion of epsins ENT1 and ENT2 (in the Δ*bem1* Δ*cla4* background) and replacing this by only a weakly expressed ENTH-domain of Ent1 (truncation). Modulating this expression should show how strong this effect is. More information on GAP interactors in this role may also be retrieved by using this mutant as the crippled starting point in an evolution experiment, akin to [22].

Furthermore, the resulting scatter of the GAPs across the other polarity pathways currently leaves room for interpretation and speculation. Given Figure 1, the components that are most shared also seem the most critical. This is most obvious when noting that actin pathway components are not just essential for polarity establishment. For example, Rho1 is needed for cell wall integrity synthesis later on during polarized growth [118,119,120]. Extrapolating, the location (wedged between pathways) of the GAPs in the Venn diagram may suggest an important (but not essential) function for each of them in establishing polarity.

Hypothetically, the GAPs might serve as an evolutionary control knob to mediate the relative hierarchy between pathways. This could be favorable in situations where different hierarchies are optimal, such as when mating is infrequent (e.g., the diploid state becomes the default), or when the bud scar is not often used (frequent sporulation). Strategic dispersal of multiple GAPs may therefore provide more handles for the cell to optimize the pathways then simply having one GAP in larger copy numbers.

### 3.3. Nrp1

After discussing two well-studied protein classes, we address an underexposed protein, namely Nrp1. With this we would like to show the difficulties and possibilities that are still open in a case when the most straightforward experiments do not provide obvious, interpretable phenotypes. Although its deletion does not have a detectable phenotype in standard lab conditions, Nrp1 is an important evolutionary [22,121]. Usually, Nrp1 is mentioned merely peripherally in articles as a bycatch in studies with an alternative focus. Therefore, a chronologically ordered literature overview does not make sense here, as very little of the research findings actually builds on previous work. An overview of the Nrp1 knowledge is given in Figure 5, where it is apparent that there are some gaps in our understanding.

Nrp1 was first described by [122], who have given the protein its name. NRP1 stands for ‘Asparagine rich protein’, the name refers to the region of the protein sequence that has many asparagines (short name: “N”). Genes are often named for their defining characteristics or functions. Reynaud and coworkers [122] did not find a phenotype or function for Nrp1, thus the seemingly nondescript name.

Nrp1 has been linked to stress response and stress granules. For example, it has been implicated in the response to glucose and oxygen [123,124,125], although the precise mechanism or function of Nrp1 in this response is not known. A hypothesis is that Nrp1 forms an aggregate or prion (like a stress granule) by its low complexity domains, because the repeated asparagine sequence in Nrp1 is often found to form prions for other proteins [126]. Nrp1 itself also seems to form a prion. In addition, Nrp1 can potentially bind and regulate mRNA. The mRNA regulation occurs through another documented domain, namely an RNA binding motif [126]. This implies that Nrp1 can bind specific mRNA sequences, however no specific mRNAs have been identified until now [127].

The supposed link between Nrp1 and the polarity network was found by authors in [22]. They showed that null mutations in *NRP1* could rescue a *bem1*Δ. This prompted interest in a search for the function of Nrp1. Another connection to the polarity network and a possible explanation for what was found in [22] is the synthetic lethality of *NRP1* with *CLA4* [128].

Diepeveen et al. [11] have found that Nrp1 is highly conserved within the Ascomycota which hints to a function of some importance. One typically expects that essential genes are the most conserved parts as they cannot easily be mutated [3]. However this is not always the case, for example *CDC42* is conserved in most fungal species (and also outside of fungi [129,130]) but it is not present in others [11]. Interestingly, *NRP1* is more conserved than several essential genes [11]. What the reason for this conservation is, is unclear. A conserved sequence does not mean conserved function or interactions. So, although a highly similar Nrp1 protein is present in a species, this does not mean it has the same function in that species.

An interesting look into the kind of experiments and research done for a protein that is similar in sequence to Nrp1 and also probably functionally related to Nrp1 is given by Whi3 [131]. Whi3 like Nrp1 has an RNA recognition motif and a low complexity/repeated region. One difference is that Whi3 has repeated glutamines and Nrp1 has repeated asparagines, though these amino acids are chemically similar. In the paper of [131] the authors removed these domains and checked functionality and localization of Whi3 in *Ashbya gossypii*. Whi3 in *A.gossypii* and budding yeast share important functionality [33,131]. Whi3 localizes to stress granules, indirectly regulates the G1/S phase transition via Cln3 and affects many mRNAs in yeast [33]. A similar role may be hypothesized for Nrp1. It also localizes to stress granules [124], affects the G1 exit [22], and may bind some mRNAs [127]. Different from Whi3, Nrp1 does not have a clear RNA target (like Cln3) that explains its functioning.

Here we will provide some research paths from different areas of research to test our hypothesis from the previous paragraph. First, from a more chemistry perspective one strategy is to look at the structure of Nrp1. The order of the domains is known, but it is unclear whether the low-complexity domain will fold, as they have been shown to be disordered [132]. This unstructured part of the protein may move about freely. It would be interesting to see if the C-terminal part of the protein does fold specifically again after the unstructured part. One would be able to visualize the structure of the protein by means of NMR [133], however it might be difficult in this case to determine the structure if it is indeed unstructured/moving.

Second, looking at Nrp1 from a cell biological perspective, deletion studies are often used. Unfortunately, this does not work as easily for *NRP1* as the single deletion does not yield any phenotype. However, in a different background a phenotype can be found, the *bem1*Δ [22]. Analyzing what happens in these cells will help understand Nrp1. This can be done on a population level by doing a fitness assay. On a single cell level one can use microscopy. A previous high-throughput study shows Nrp1 present in the cytoplasm [103]. A more detailed study focusing on polarity establishment can give more insight into the functioning of Nrp1, especially when combined with the previous approach (different genetic backgrounds).

By the same token, a more in-depth analysis of the RNA binding ability of Nrp1 is also relevant. Again, using Nrp1 in different environments may give different results and find different specific RNAs that are bound. RNA chip-seq has become easier to execute over the last few years [134] and thus now it might be possible to do the proposed experiments.

The ultimate goal is to find a molecular mechanism for Nrp1, but this goal cannot be reached without knowledge of other aspects of the network. Nrp1 is a good example of a protein that is buried deep in the network, which makes the investigation challenging. However, it is worthwhile to dig deep and find the hidden functionality of proteins like Nrp1 that seem neutral at first glance, but have a significant evolutionary role.

## 4. Outlook

Protein−protein interaction networks have diverse properties that influence its evolution, such as redundancy, hierarchy and neutrality. We have advocated that studying yeast polarity provides a suitable starting point for general studies on the evolution of these complex networks, as it exhibits many of the aforementioned traits and includes the evolutionary relevant spatiotemporal dynamics [17]. From work in [26], it had become clear that predicting epistasis, which precedes predicting evolution, required deep understanding of the interactions and reactions constituting a protein network like yeast polarity.

However, our case studies demonstrate that obtaining this knowledge in a complex network architecture is far from trivial due to the aforementioned redundancy, hierarchy and neutrality. While some genes generate obvious phenotypes from which the roles of the corresponding gene products are easily deduced, for others the required information is only revealed after multiple rounds of precisely designed experiments. The latter situation can be due to various reasons related to the network architecture, for example the associated protein is found deep down in the hierarchy, forms part of a redundancy in the network or is currently otherwise peripheral to the core function. Regardless of the origin, encountered neutrality is a complication, which is a particularly unresolved feature in the third case study (Nrp1).

By exploring the past and present of the polarity network, we have aimed to determine how to advance for future research, to answer the open questions in the field and underline research opportunities. Although the yeast polarity network has been studied for many years, there still remain many unsolved mysteries. For example, the study by [22] gave much insight into the evolution and possible back-up mechanisms of the polarity network, but gave rise to the question of how Nrp1 is involved.

Much research has been done under perfect lab conditions, which results in specific results. It would be interesting to also design experiments that explore the genotype−phenotype map in different conditions. This would make it possible to find previously hidden components, that do not show up under standard lab conditions. Another possibility is changing the environment together with the expression level of specific genes, which can affect fitness [135], as has been shown for Cdc42 [23].

In vitro work is also an important next step to isolate parts of the network and see if these parts can independently perform a function [136]. As an example in budding yeast, in vitro reconstitution of a She-protein mediated mechanism of asymmetric mRNA transport revealed subtle details that were hard to demonstrate in vivo [137].

From an evolution standpoint it is interesting to see how the network as it is in current strains came to be and what the variation is that can be found in the wild. The variation within a wild population shows the spread that is available in the genotype map. It provides insight into what genotypes are preferable in certain environments. Apart from the genotype showing the history of a population, also, for example, the expression levels of proteins can be inherited [138]. Such epigenetic inheritance can be important for the reaction of an organism to stress and other environmental factors [139].

The yeast polarity examples discussed also delineate a general route forward in dismantling complexly connected protein networks. A graphical overview is depicted in Figure 6. The arrow heads indicate the type of information obtained from proteins inside the network, and higher degrees of information are successively more difficult to obtain, and usually rely on first reaching the previous level. In this way, we work our way deeper into the protein network and slowly but steadily elucidate beyond the obvious phenotypes.

In the top category, much work has been done determining the effects of simple deletions. For budding yeast, a large knock-out database has been present for more than two decades [12]. The ease of experimentally parallelizing the deletion construction even allows for ample double deletion data constituting genetic interactions, which for multiple model systems is bundled in the BioGRID database [140]. Yet, there is a myriad of combinatorial possibilities for gene deletions, and it has become clear from the GAP and Nrp1 examples that these need to be explored intelligently, rather than by brute force. Well-designed starting points for evolution experiments, for example to find that GAPs and Nrp1 genetically interact with Bem1 in [22], or SATAY assays [141] can elucidate interactions of genes of interest with important domains that only surface with strong phenotypes in the right genetic background.

However, genetic interactions can be very indirect. Epistasis is known to act globally even on unrelated networks during adaptation [142], so gathering physical information is a welcome next step. The efficient two-hybrid screens date back more than three decades [143], where two proteins fused to a DNA binding domain and transcriptional activator, respectively, promote transcription of a reporter gene when interacting. To follow interactions across the cell (see also next paragraph), FRET imaging, where two fluorescent fusion proteins cause the emission spectrum to shift when in close proximity, has also been extensively used [144], but this method also relies on the proteins of interest to be tolerant to protein fusions. More subtle modifications for tagging are used for co-immunoprecipitation [145], and are still heavily used in budding yeast [146]. Once physical interactors have been established, these can be further confirmed with their relevant binding sites by point mutation to influence the binding, as in, e.g., [84] for Bem1. This level of precision also means a move towards low-throughput data gathering, but can be useful to conjecture how cells after the deletion of *BEM1* evolved [23].

While the physical interactions constitute a rudimentary form of the protein network, the next step in understanding originates from adding spatiotemporal information. For example, Bud3 is a GEF for Cdc42, but only during early G1 phase [105], before the timing pathway gives the cue for symmetry breaking. If the function of interest is symmetry breaking, this can be excluded from the network overview as in Figure 1. Generally, a high-throughput manner for establishing protein localization in vivo has been with fluorescent protein fusions, particularly with GFP variants [103]. As aforementioned, immunostaining provides a similar option for tagging and hence localization, but requires fixation of the cells, making the temporally transient contributions of components more difficult to trace. If temporal rather than spatial information on the importance of a protein is of the essence, ingenious solutions exist that conditionally disable the protein of interest, after which the results can be swiftly observed. Examples include degron systems [147,148] and optogenetic tools [149]. Finally, as with Nrp1 localization of mRNAs may be the important function, options to trace these are also present with in situ hybridization, as described in, e.g., [150].

Ultimately, when all previous steps have been performed, it becomes possible to make the next step towards complete understanding, which would be mechanistic understanding. As shown in the case of actin, if insufficient information is available, modelling can lead to uncertain and contradictory results, such as is the case with the role of actin in polarity establishment [24,95,96]. If possible, the most unambiguous results would come from bottom-up approaches, such as modelling Michaelis−Menten kinetics for metabolism (e.g., [151] (pp. 165–180)) or solving reaction−diffusion systems, as for polarity done in [23,87] in case proteins cannot be assumed to be uniformly distributed. The more complete understanding of the protein network is then put to the test by the prediction of observed adaptive trajectories in historical or experimental evolution, as done, for example, in the latter case for the path of [22] in [23,26]. Considering the many options and solutions evolution can explore, accurate prediction of evolution is the best guarantee that the full extent of the knowledge of a network has been explored.

In summary, the general path to full network understanding as outlined in Figure 6 brings us from genetic and physical interactions to visualizing precise protein dynamics, modeling and full reconstitution. While initially, even the genetic interaction map was a tedious chore, high-throughput studies and bioinformatics tools continue to facilitate the gathering of information. Considering the speed with which the technological advances occur, the necessary data for network understanding becomes feasible for many more functions and organisms. This marks the relevance of establishing a generalizable and efficient workflow to obtain the right network data and use it for understanding and predicting its evolution.

## Figures and Tables

**Figure 1 cells-09-02534-f001:**
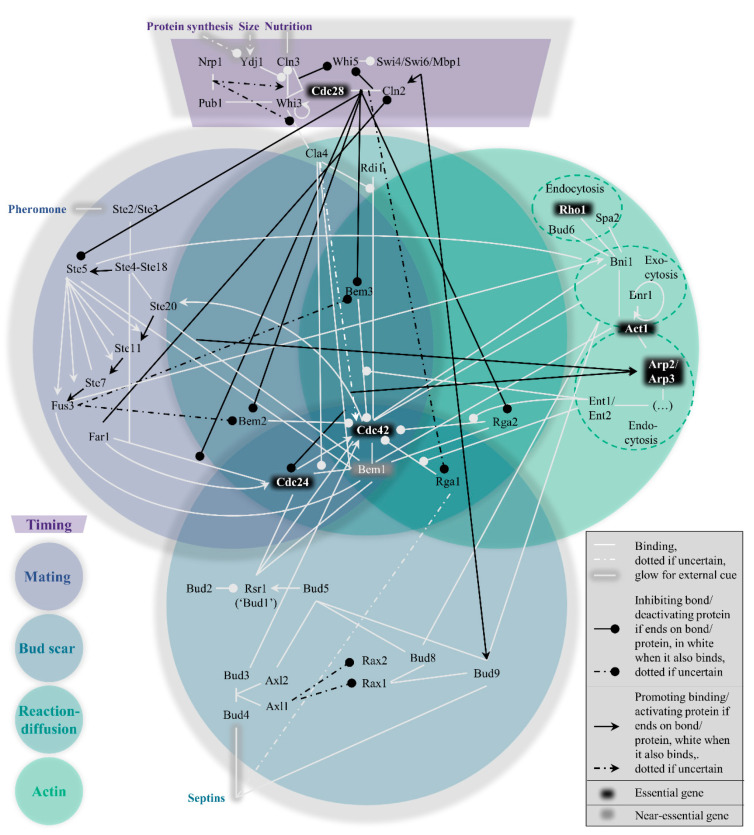
Venn diagram representation of the hierarchical, nonmodular pathway structure for protein−protein interactions in budding yeast polarity establishment (Table A1 for references).

**Figure 2 cells-09-02534-f002:**
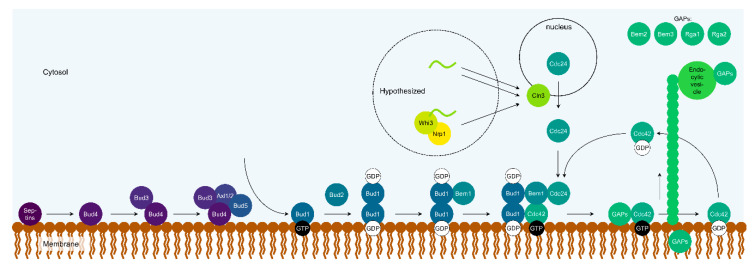
Mechanistic model for the haploid bud scar pathway, the reaction−diffusion pathway including the GAPs, and the hypothesized role of Nrp1 in the timing pathway.

**Figure 3 cells-09-02534-f003:**
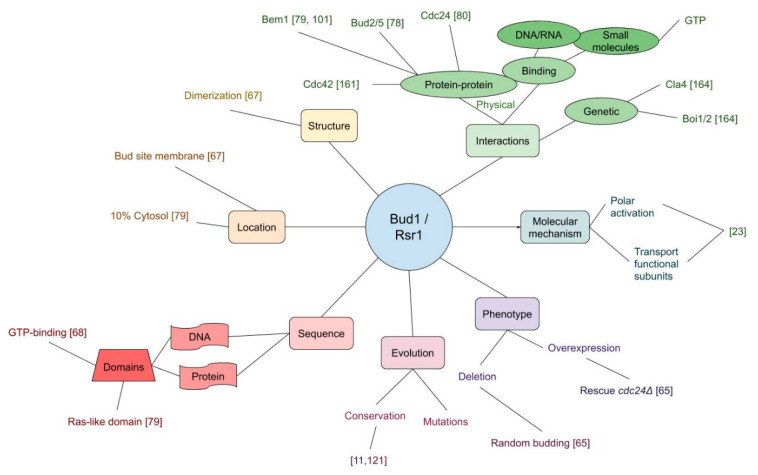
Mind map representation of the bud scar protein Bud1 summarizing its most important properties from literature, such as localization, domains and interactions.

**Figure 4 cells-09-02534-f004:**
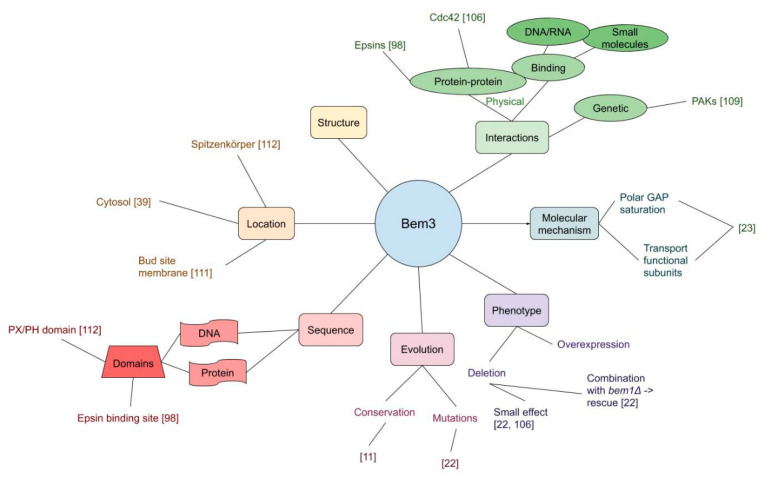
Mind map representation of the GTPase activating protein (GAP) Bem3 summarizing its most important properties from literature, such as localization, domains and interactions.

**Figure 5 cells-09-02534-f005:**
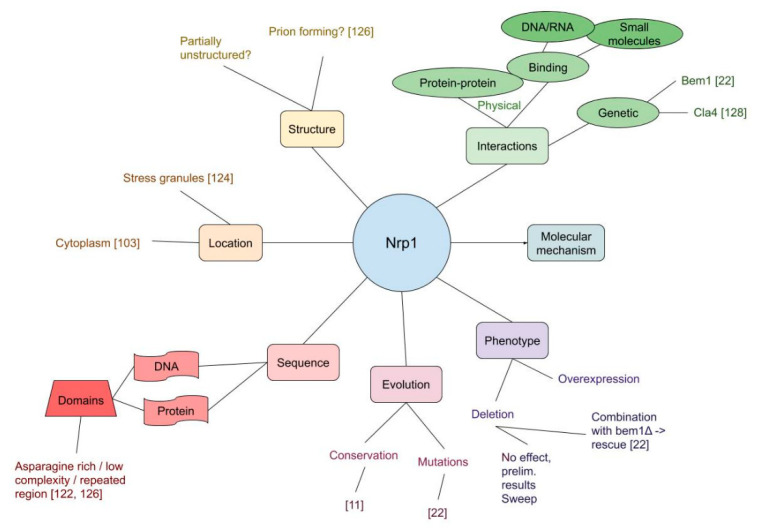
Mind map representation of Nrp1 summarizing its most important properties from literature, such as localization, domains and interactions.

**Figure 6 cells-09-02534-f006:**
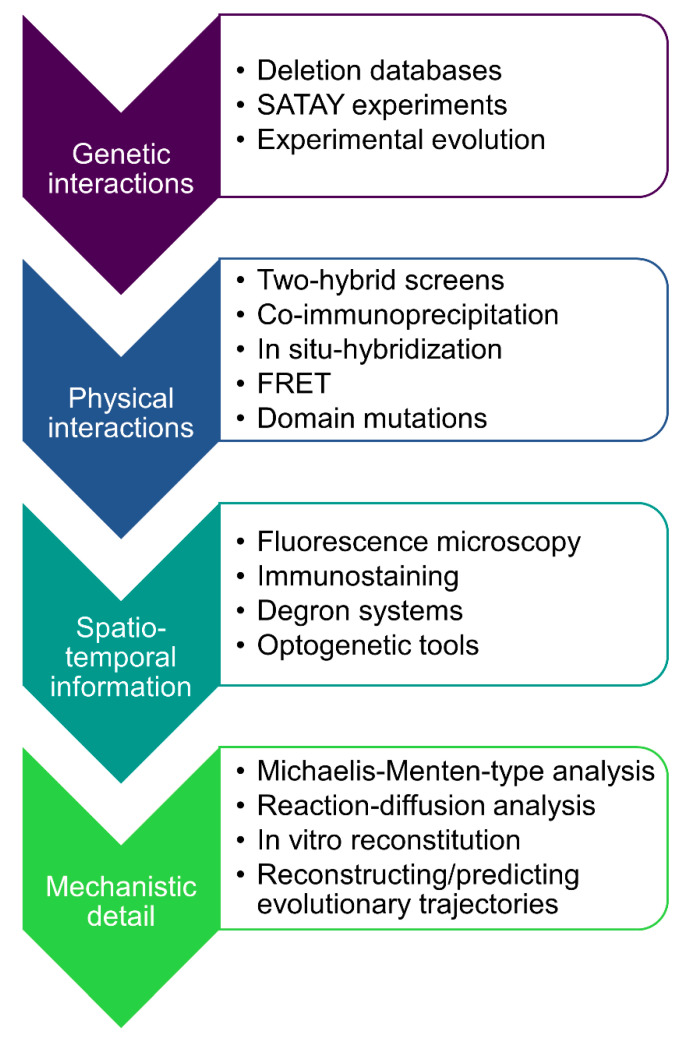
Flow chart for understanding the roles of proteins inside a complex network. Level of detail increases from top to bottom, while the enumerations of accompanying experiments corresponding to a level are ordered in decreasing feasibility of high-throughput data generation.

**Table 1 cells-09-02534-t001:** Definitions of important terms used throughout the introduction.

Term	Definition
Protein network	Group of proteins with physical interactions together performing a function
Connectivity	Degree to which parts of the network are embedded with other parts in the network. In this sense, it can be received as the reciprocal of modularity.
Modularity	Potential to group parts of a protein network given a certain representation of the protein network (e.g., in terms of mechanisms, genetic or physical interactions)
Hub protein	Highly connected protein in a network (often essential)
Neutrality	No consequence of a mutation to phenotype (in current environment)
Hierarchy	Clear layering of pathways inside a protein network
Redundancy	Multiple mechanisms that can to some extent interchangeably contribute to the same function

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
