# Peer review of "The Path towards Predicting Evolution as Illustrated in Yeast Cell Polarity"

_cells, 2020, doi:10.3390/cells9122534_

Round 1
Reviewer 1 Report
How to understand cells well enough to predict evolution: polarity establishment in budding yeast as a case study.
Werner Karl-Gustav Daalman , Els Sweep and Liedewij Laan
In order to show that the cellular network architecture impacts the evolution possibilities, the authors introduced that polarity network in S. cerevisiae serves as a great case study. Five polarity pathways are concisely described, namely the timing clue, mating pathway, bud scar signal pathway, reaction-diffusion and actin pathway. The interaction between these pathways is briefly discussed. As case studies, bud scar protein research history was described, GAPs protein was reviewed and Nrp1 research status was discussed in details. The last section discussed the outlook of future research direction for polarity pathway and mainly about Nrp1 gene.
Main points:
- This review is mainly about polarity pathway and proteins involved, only the first section and the fourth section briefly discussed network properties. It is not at all clear that how polarity establishment can be used as a case study to understand cell well enough to predict evolution. It is more like a review just for polarity. From the title and the introduction, readers expect taking polarity as an example, how the change of cellular architecture interacts with evolution. However, the authors only discussed the research review about the protein involved in polarization.
- The authors should either add research reviews about how network changes affects evolution in polarization pathways, or completely change the paper into a polarity pathway review.
- The title did not reflect the content of the review body. To fit the title, the authors should review more network related literature or discuss the possibility of using polarity as a case study to infer the whole PPI network property. Where are the similarities between cell polarity and the whole PPI network features, and what cell polarity network property infers?
- The case study section 3.1 seemed like a more detailed review than section 2.3. There is very few case study about network involved.
Minor points:
- “ For example, hubs, proteins with many 33 binding partners, tend to mutate slower.” The hub proteins probably mutate at the same rates as other proteins but the mutated ones cannot adapt. The better wording should be “evolve slower”.
- Too many “Error! Reference source not found”, the author should address this before submission.
- There are two “3.2 ” sections. Nrp1 should be 3.3.
Author Response
We thank the reviewers for their useful comments that helped us to improve our manuscript. In addition, we welcome the notification concerning the inclusion of the paper of Goryachev and Pokhilko FEBS Let. 2008 regarding theoretical work performed on the reaction-diffusion part of yeast polarity. While initially we had only referred to the most up-to-date model concerning the Bem1-mediated positive feedback loop, we understand the value of the context of historical evolution of theoretical work dating from before the FEBS paper. Therefore, we addressed this in the document.
The first reviewer has made the following four main points, that involve important concerns we wish to resolve in order to clarify the points in our review. We adhere to the line numbering in Word with all markup changes visible on.
- This review is mainly about polarity pathway and proteins involved, only the first section and the fourth section briefly discussed network properties. It is not at all clear that how polarity establishment can be used as a case study to understand cell well enough to predict evolution. It is more like a review just for polarity. From the title and the introduction, readers expect taking polarity as an example, how the change of cellular architecture interacts with evolution. However, the authors only discussed the research review about the protein involved in polarization.
Based on this comment we improved the link showing how understanding of the polarity network facilitates prediction of evolution. Very recently, we posted a preprint on Biorxiv concerning this missing link, which we can now include to make the point to be expected from the title of this review more apparent. The text in lines 98-112 and lines 480-495 has been added, to include a practical application of how the detailed network understanding. We propose that this understanding can result in the construction of accurate predictions for epistasis, and subsequently, for evolution.
- The authors should either add research reviews about how network changes affects evolution in polarization pathways, or completely change the paper into a polarity pathway review.
While it would be very valuable to include literature on how network changes affect polarity evolution, there are not many concrete predictions or descriptions other than those in papers already mentioned in lines 82-90. This reaches the core of the problem we propose, namely that without detailed network information, predicting evolution is difficult. What is typically possible is to resort to more conventional links between network architecture and general evolutionary relevant terms, such as feedback motifs leading to robustness. Examples have now been included in lines 91-96, where the need for more specific predictions has been stressed.
Furthermore, we realize the ability to interpret or even predict evolutionary trajectories based on the network interactions and mechanisms is a desirable goal and an ultimate test for network understanding. We added this explicitly to the flow chart in fig. 6 and included this in the text, where we discuss this figure in lines 574-577.
- The title did not reflect the content of the review body. To fit the title, the authors should review more network related literature or discuss the possibility of using polarity as a case study to infer the whole PPI network property. Where are the similarities between cell polarity and the whole PPI network features, and what cell polarity network property infers?
Our interpretation is that we are encouraged to more thoroughly discuss the differences and similarities between yeast polarity and general networks. We make the argument that yeast polarity has many of the desirable network properties to be representative, with the exception of containing much gene regulation, which is covered in other templates, as now discussed in lines 59-81.
- The case study section 3.1 seemed like a more detailed review than section 2.3. There is very few case study about network involved.
We understand this comment as a request to specify more concretely the purpose of section 3.1 The intend is not a redundancy with section 2.3, because whereas 2.3 concerns the brief cell biology description of the whole bud scar pathway, 3.1 revolves around how this knowledge was obtained, also in historical context, and only for a subset of proteins. Section 3.1 helps to generate a path towards understanding in a more complex case study (as in 3.3). Because the bud scar knowledge is so far advanced, the purpose is the use of the case study as an example to work towards in the other two cases, rather than requiring much future work itself. We therefore included lines 269-275 to stress this point and resolve the apparent redundancy.
The first reviewer also suggested minor points, which were promptly addressed as follows:
- “ For example, hubs, proteins with many 33 binding partners, tend to mutate slower.” The hub proteins probably mutate at the same rates as other proteins but the mutated ones cannot adapt. The better wording should be “evolve slower”.
We agree, and “tend to mutate slower” is replaced by “tend to evolve slower”.
- Too many “Error! Reference source not found”, the author should address this before submission.
These broken references have been substituted by the literal texts corresponding to the respective referred objects.
- There are two “3.2 ” sections. Nrp1 should be 3.3.
We corrected the section number to 3.3.
Reviewer 2 Report
In “How to understand cells well enough to predict evolution: polarity establishment in budding yeast as a case study,” Daalman et al. review the networks underlying polarity establishment in budding yeast as a model to understand how one can use cell-biological knowledge to predict evolution. They review five networks central to polarity establishment and discuss both intra-network and inter-network interactions. They highlight examples of modularity and redundancy, as these factors are thought to shape network evolution. They next highlight how knowledge concerning the functions of three proteins in these polarity networks was elucidated, with a focus on the role of evolution in this discovery process. They end by discussing potential future work on polarity establishment and its connection to evolution.
This review did an excellent job of mapping concepts from evolutionary biology to yeast cell biology, especially those concerning evolution of network structure, and in comprehensively covering the details of polarity establishment. The one area where I think the manuscript can be improved is in regards to the idea of predicting evolution, as mentioned in the title and abstract. I agree with the authors that establishing complex models of cells is essential to predicting their evolution. They did a thorough job of covering how network models of polarity establishment have been constructed. But after reading the manuscript in its current form, I’m still not sure how well these network models allow us to predict their evolution.
It would be beneficial to specifically discuss what knowledge is necessary for prediction in this system and to estimate how close the field is to gaining this knowledge to accurately predict evolution. Previous experimental evolution and phylogenetic studies with this system suggests we are approaching this goal, but I would like the authors’ thoughts on how close we are. Most papers on predicting evolution are vague (and perhaps necessarily so). By providing some idea of what is needed to make accurate predictions in this system, this review would stand out from other studies on predicting evolution.
Minor comments/suggestions
Line 51: “)” missing after [12]
Line 110: “;” should be “:”, I believe
Line 244: “intelligent point mutations”. Do the authors mean “intelligently-chosen point mutations” or” specific point mutations chosen by the experimenter” here?
Lines 266 & 339: Both of these sections are “3.2”
Line 329: “…polarity establishment, for example, Rho1…” should be two sentences, i.e., “…polarity establishment. For example, Rho1…”
Line 391: I don’t think parentheses are needed around the citation here.
Line 423: “as s has” I think the “s” should be deleted.
Author Response
We thank the second reviewer for his/her compliments and agree with the important point noted. To our understanding, we can summarize the main point of the second reviewer as follows: we insufficiently describe how network models lead to prediction of evolution, i.e., what is needed to reach this goal.
Based on this comment we improved the link showing how understanding of the polarity network facilitates prediction of evolution. Very recently, we posted a preprint on Biorxiv concerning this missing link, which we can now include to make the point to be expected from the title of this review more apparent. The text in lines 98-112 and lines 480-495 has been added, to include a practical application of the detailed network understanding. We propose that this understanding can result in the construction of accurate predictions for epistasis, and subsequently, for evolution.
The second reviewer also suggested minor points, which were promptly addressed as follows:
- Line 51: “)” missing after [12]
We added “)” as suggested.
- Line 110: “;” should be “:”, I believe
We replace “:” by “;”as suggested.
- Line 244: “intelligent point mutations”. Do the authors mean “intelligently-chosen point mutations” or” specific point mutations chosen by the experimenter” here?
We replaced “intelligent” by “intelligently-chosen”.
- Lines 266 & 339: Both of these sections are “3.2”
As mentioned by the same comment for the first reviewer, we corrected the second 3.2 to section 3.3.
- Line 329: “…polarity establishment, for example, Rho1…” should be two sentences, i.e., “…polarity establishment. For example, Rho1…”
We have split the sentence after “establishment”.
- Line 391: I don’t think parentheses are needed around the citation here.
We have removed the parentheses.
- Line 423: “as s has” I think the “s” should be deleted.
We have deleted the “s”.
Finally, we made minor revisions to the manuscript unrelated to the commentary by the reviewers. Aside from typographical corrections, this entailed the realization that recruitment of Bem1 (and only then Cdc24) by Bud1/Rsr1-GDP to the bud site seems more in line with literature, which required updating figure 1 and 2, and lines 215-216, 300-302 and two supplementary table entries accordingly.
We hope to have fully and adequately addressed the points raised by this reviewer.